# Application of Model Predictive Control for Large-Scale Inverted Siphon in Water Distribution System in the Case of Emergency Operation

**Zheli Zhu** [1] , **Guanghua Guan** [1] , **Zhonghao Mao** [1] , **Kang Wang** [1],*  , **Shixiang Gu** [2] **and Gang Chen** [2]

[1]  School of Water Resources and Hydropower Engineering Science, Department of Agricultural Water Resources Engineering, Wuhan University, Wuhan 430072, China; 2014301580021@whu.edu.cn (Z.Z.); GGH@whu.edu.cn (G.G.); 2011301580373@whu.edu.cn (Z.M.)

[2]  Yunnan Survey and Design Institute of Water Conservancy and Hydropower, Kunming 650021, China; gushixiang@ynwdi.com (S.G.); chengang@ynwdi.com (G.C.)

*  Correspondence: wwangkang@whu.edu.cn

**Abstract:** The emergency control of Menglou~Qifang inverted siphon, which is about 72 km long, is the key to the safety of the Northern Hubei Water Transfer Project. Given the complicated layout of this project, traditional emergency control method has been challenged with the fast hydraulic transient characteristics of pressurized flow. This paper describes the application of model predictive control (MPC), a popular automatic control algorithm advanced in explicitly accounting for various constraints and optimizing control operation, in emergency condition. For the fast prediction to the pipe-canal combination system, a linear model for large-scale inverted siphon proposed by the latest research and the integrator-delay (ID) model for open canals are used. Simulation results show that the proposed MPC algorithm has promising performance on guaranteeing the safety of the project when there are sudden flow obstruction incidents of varying degrees downstream. Compared with control groups, the peak pressure can be reduced by 17.2 m by MPC under the most critical scenario, albeit with more complicated gates operations and more water release (up to $9.75 \times 10^4$ m$^3$). Based on the linear model for long inverted siphon, this work highlights the applicability of MPC in the emergency control of large-scale pipe-canal combination system.

**Keywords:** model predictive control; emergency control; linear model; Northern Hubei water transfer project; long inverted siphon; pipe-canal combination system

## 1. Introduction

The Menglou~Qifang inverted siphon is an important part of the Northern Hubei Water Transfer Project (NHWTP), which is about 72 km long and has a design flow of 38 m$^3$/s. It is rare in the worldwide water transfer projects, so there may be no referable experience about the emergency control, which is vital to the safety of the NHWTP. When there is an accident, untimely or unreasonable control measures may lead to serious secondary accidents, such as tube bursting caused by water hammer, overtopping of open canals caused by too fast gate operations, and so on. Therefore, it is of great significance for the NHWTP to study the hydraulic response and emergency control of the ultra-long inverted siphon in accident conditions.

The most conventional method for such a problem is making an emergency dispatching plan, which can be divided into three steps. The first step is to give guidelines for emergency dispatch based on specific project and experience [1,2]. Then, a variety of gates operation schedules are set out to study the hydraulic response [3]. Finally, the optimal group is selected to develop the emergency

scheduling plan according to the simulation results [4–6]. The biggest problem about the emergency dispatching plan is that it is only applicable to typical accidents in typical canals and may consume a lot of time and computing resources. Moreover, if there is an accident condition that is not considered in the plan, the performance of emergency control may still largely depend on the operators' personal experience and proficiency.

So canal automation methods are needed in emergency control [7]. Soler et al. [8] used the "Gómez, Rodellar and Soler" (GoRoSo) feedforward algorithm to compute the gate trajectories that could smoothly carry the canal from the initial to the final state by keeping the water depth constant at checkpoints in the case of an emergency by closing the upstream pool. Lian et al. [9] and Xu et al. [10] proposed the calculation formula of emergency gate falling time in sudden water pollution accidents to control the pollution in the channel of accident. However, they have only focused on the transfer range of pollutants, not the safety of the canals system structures. Kong et al. [11] used proportional integral (PI) water level difference feedback control algorithm to prolong the continuous delivery time of pools with offtake delivery demand under sudden upstream water interruption. Cui et al. [7] and Kong [12] proposed a two-step control algorithm under emergency conditions to deal with the recovery characteristics of canals, respectively. In the meantime, the effect of gate controlling on canals and stable water diversion in upstream reaches are also taken into account.

In these studies, the research object of emergency dispatching is the open water system, rather than the pipe-canal combination system in which the hydraulic transition process of pressurized flow is faster and more intense than that of open flow, and slight operating changes may cause large differences in results which make the conventional method of making an emergency dispatching plan unsuitable. It would be applicable if the pressure fluctuations in the inverted siphon can be predicted and gates actions are taken in advance according to the safety restrictions. At this time, model predictive control (MPC) is a good choice. The biggest advantage of MPC is the ability to explicitly account for constraints on water head, gate movements and so on, which is hard for other automatic control methods, such as PI controllers and linear quadratic regulators (LQR) [13–15]. MPC has a wide range of applications, in addition to conventional water dispatching [16–18], there are also unconventional operation conditions, such as risk mitigation [19], drought [20–22] and flood [23,24] or even emergency conditions. Vierstra [25] dealt with an unexpected failure of a pump station in the South–North Water Transfer Project, which shows MPC can anticipate the hydraulic interaction between all canal pools and supply as much water as possible under the structure failure. Other studies about emergency controls by MPC are mostly in other fields, such as electric control [26,27] and unmanned autonomous vehicles [28,29].

In general, MPC is used in the open canal system, rather than the pipe-canal combination system. The main reason is that there are mature linear models for open canal flow as the internal progress model of MPC, such as integrator delay (ID) model, integrator-delay zero (IDZ) and reduced Saint-Venant models [30], but the pressurized flow does not. The authors [31] recently proposed a linear model that relates the pressure head variations at the downstream end of an inverted siphon to the flow rate variations at two ends. Therefore, this research is carried out to evaluate the predictive effect of the linear model proposed by Mao et al. [31] as the internal model of MPC on the one hand, on the other hand to study the applicability of MPC in emergency control by considering security constraints and preventing secondary accidents. The structure of this paper is as follows. Firstly, the project is introduced in the Section 2 for its particularity. In Section 3, the principles of simulation and model predictive control are presented. Then, in Section 4, four test scenarios of different accident levels and their control groups are simulated and analyzed to evaluate the effect of MPC on emergency control of the NHWTP. Discussion and conclusions are drawn in the last two sections.

## 2. The Northern Hubei Water Transfer Project

The Northern Hubei Water Transfer Project draws water from Danjiangkou Reservoir, and its Menglou~Qifang inverted siphon (i.e., Pool 2 in Figure 1) is 72 km long, which is rare in the worldwide

water transfer projects. During the hydraulic transition, the unsteady flow in the open canals and in the pipelines will influence each other. In order to study the hydraulic response of Menglou~Qifang inverted siphon under accident conditions, the modeling scope in this study should cover the long inverted siphon and the open canals at both ends (i.e., Pool 1 and Pool 3 in Figure 1). The initial flow rate is the design flow rate, 38 m³/s, and there are no offtakes along the line. The layout is shown in Figure 1 and basic parameters are presented in Table 1. The initial openings of the gates on open canals are set as 80% of the maximum opening according to experience. The opening and closing speed of the gates is limited to 0.5 m/min.

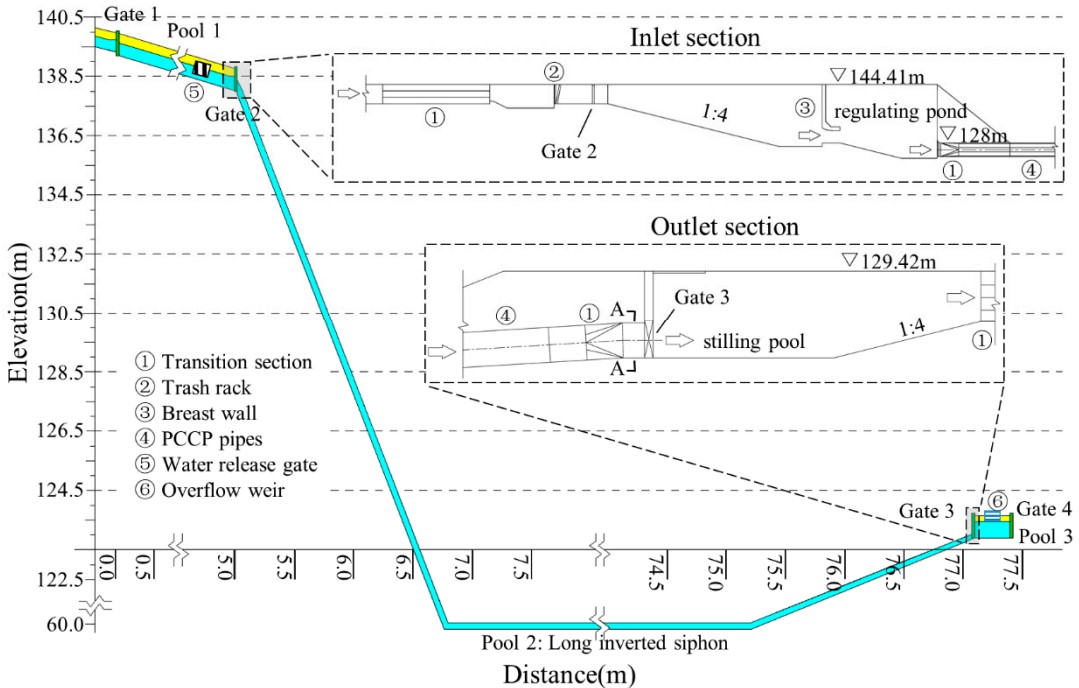

**Figure 1.** Layout of the case canal segment.

**Table 1.** Basic parameters of each pool.

| Pool | Pool Length (km) | Gate Width (m) | Maximum Opening of Gate (m) | Gate Initial Openings (m) | Start Elevation (m) | End Elevation (m) | Downstream Initial Flows (m³/s) | Downstream Initial Level (m) |
|---|---|---|---|---|---|---|---|---|
| Heading | | 7.4 | 4.9 | 3.92 | 140.3 | | | 144.36 |
| 1 | 5.32 | 5 × 3 | 4.8 | 3.84 | 140.2 | 138.99 | 38 | 143.41 |
| 2 | 72.08 | 3.8 × 3 | 3.8 | 3.8 | 124.2 | 119.9 | | 128.42 |
| 3 | 0.32 | 8 | 4.8 | 3.84 | 123.9 | 123.89 | | 128.35 |

On Pool 1, a water release gate is arranged 100 m away from the inlet section of the inverted siphon, with a width of 3 m, a height of 4.8 m and a designed discharge of 38 m³/s. An overflow weir is set on Pool 3, which is 200 m away from the outlet section of the inverted siphon. The width of overflow weir crest is 60 m and overflow will begin when the water level is higher than 128.50 m. Pool 2 contains a long inverted siphon, which can be divided into three parts: inlet section, pipe body section and outlet section. Three DN3800 mm pre-stressed concrete cylinder pipes (PCCP pipes for short) are used in the pipe body section, and the maximum design pressure head of outlet pipeline is 61 m. The inlet section is equipped with a two-stage free drop section, and connected with a regulating pond through the breast wall, with a top elevation of 144.41 m. Gate 3, the control gate of the outlet section, is located at the junction of the pressure flow in the pipeline and the open flow in Pool 3.

## 3. Model Formulation

In this part, the control purposes, the methodologies about simulation and automatic emergency control, and the scenarios designed for simulation will be described. MPC will be applied to two objects: the exit gate of the long inverted siphon (Gate 3) and the water release gate on Pool 1 to prevent pipes from bursting and open canal from overtopping, respectively. Four test scenarios are given to test the emergency control effect of MPC when there are sudden flow obstruction incidents of varying degrees downstream. Variations used throughout this paper are listed in Appendix A.

### 3.1. Control Purposes

Under sudden accidents, the following safety conditions need to be met by reasonable emergency control strategies in the progress of gate closing from initial opening to target opening.

1.　The cross section in front of Gate 3 (Section A-A in Figure 1) is easy to burst when the pressure exceeds the limit in the progress of fast closing, so the internal water pressure of this cross section must be controlled below 61 m.
2.　The water level in the regulating pond should be controlled between 128 m and 144.41 m to prevent overflow or air intake at the inlet section.
3.　To ensure that the overtopping accident will not occur in the open canals during the control process. When the water level in Pool 3 exceeds 128.50 m, it will automatically overflow because of the overflow weir. According to experience, the possibility of overtopping in Pool 3 is small. Nevertheless, the water level in Pool 1 needs to be reasonably controlled by the water release gate to prevent overtopping.

### 3.2. Preissmann Slot Method

The Preissmann slot method (PSM) has been widely used in modeling transitions between free-surface flow and pressurized flow [32]. Comparing the continuity equation and motion simulation of unsteady flow in open canals and pressurized flow in pipes, as shown in Equations (1) and (2), respectively, it can be seen that if a narrow slot is assumed on the top of pressurized flow (as shown in Figure 2) and the width of slot is set as Equation (3), the control equations of the two flow regimes will be the same. Therefore, Equation (1) can be used as the basic equations to uniformly describe the open flow and the pressurized flow.

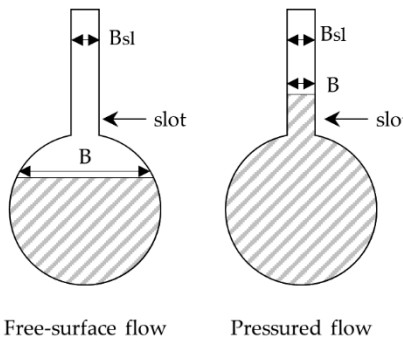

**Figure 2.** The Preissmann slot method.

At this time, the variable $H$ represents the water depth in the open flow, or the pressure head acting on the bottom of the pipe in the pressurized flow. Further, the variable $B$ represents the width of the water surface in the open flow, or the width of slot in the pressurized flow. Since the width of the narrow slot is very small, the influence on the wetted area and hydraulic radius can be neglected, but the transmission of pressure waves in the pipeline can be simulated by a suitable slot size.

$$\begin{cases} \frac{\partial H}{\partial t} + v\frac{\partial H}{\partial x} + \frac{A}{B}\frac{\partial v}{\partial x} = 0 \\ gA\frac{\partial H}{\partial x} + v\frac{\partial v}{\partial x} + \frac{\partial v}{\partial t} = g(S_0 - S_f) \end{cases} \tag{1}$$

$$\begin{cases} \frac{\partial H}{\partial t} + v\frac{\partial H}{\partial x} + \frac{a^2}{g}\frac{\partial v}{\partial x} = 0 \\ gA\frac{\partial H}{\partial x} + v\frac{\partial v}{\partial x} + \frac{\partial v}{\partial t} = g(S_0 - S_f) \end{cases} \tag{2}$$

$$B_{sl} = \frac{gA}{a^2} \tag{3}$$

where $H$ is the water depth in the open flow, or pressure head in the pressurized flow (m); $A$ is the wetted area (m$^2$); $B$ is the width of water surface or slot (m); $B_{sl}$ is the width of the narrow slot (m); $g$ is the acceleration of gravity (m/s$^2$); $a$ is the speed of the acoustic wave, which is taken as 1054 m/s; $v$ is the flow velocity (m/s); $S_f = n^2v^2/R^{4/3}$ is the hydraulic slope; $n$ is the Manning coefficient; $R$ is the hydraulic radius (m); $S_b$ is the bed slope.

### 3.3. Model Predictive Control

MPC is capable of foreseeing delivery problems and dealing with various constraints on the controlled variable (water level or pressure head) or the control variable (gate position), which can ensure safe hydraulic transition by advance prediction and objective function optimization. In this paper, the water depth in Pool 1 and the pressure head before Gate 3 are both important controlled variables, directly related to whether there will be secondary accidents such as overtopping or tube-burst in the emergency control progress. Accordingly, MPC will be implemented on the Gate 3 and the water release gate, which can be called MPC-1 and MPC-2, respectively.

MPC-1 is configured with four aspects: internal model, receding horizon, constraints and objective function [14], which will be explained below. It is activated at the moment of the accident. Every control cycle, ten control action options will be predicted by Mao's linear models [31] and compared by the constraints and objective function. At the end of the cycle, only the most optimal gate opening increment of Gate 3 is chosen as the output. MPC-2 is consistent in the internal model and receding horizon with MPC-1, but it controls the water release gate only based on the predicted results of every control cycle without constraints and optimization. More details of the control strategies are shown in Table 2. The schematic overview of the control system can be seen in Figure 3. The parameter C is set to prevent the inverted siphon intake, which will be explained in Section 3.4. Further, the actual water system is simulated by solving the Saint-Venant equation.

**Table 2.** Details of the control strategies.

| Parameters | Description |
|---|---|
| **MPC settings** | |
| controlled variable | water level in Pool 1 and pressure head in the inverted siphon |
| control variable | gate opening of the water release gate and Gate 3 |
| the internal model for the open canals | the ID model |
| the internal model for the inverted siphon | the linear model recently proposed by Mao et al. |
| the control interval | 10 time steps |
| the prediction horizon | 10 time steps |
| the opening and closing speed of the gate | 0.5 m/min |
| computation time step | 1 min |
| the dead-band of each gate movement | 5 cm |
| **MPC-1 for the exit gate of the long inverted siphon** | |
| the controlled gate | Gate 3 |
| the time when MPC-1 is activated | the moment of the accident |
| the time when MPC-1 is turned off | the moment when the safety is guaranteed and target opening is reached |
| the control action options of gate opening increments | [−1.0 m, −0.75 m, −0.5 m, −0.25 m, 0 m, 0.5 m, 1 m, 1.5 m, 2.0 m and 2.5 m] [1] |
| the constraints | Equation (7) |
| the objective function | Equations (8)–(10) |

**Table 2.** *Cont.*

| Parameters | Description |
|---|---|
| **MPC-2 for the water release gate** | |
| the controlled gate | the water release gate |
| the time when MPC-2 is activated | the moment of the accident |
| the time when MPC-2 is turned off | \ |
| the constraints | \ |
| the objective function | \ |
| the decision of the gate opening by predicting | Equation (11) |
| **Other control strategies** | |
| the opening of Gate 1 and Gate 4 | the gates operation schedule |
| the opening of Gate 2 | the inverse calculation module for a given target flow |
| the control interval for Gate 2 opening adjustment | 15 time steps |
| the opening of Gate 3 after MPC-1 is turned off | the inverse calculation module for a given target flow |
| the control interval for Gate 3 opening adjustment | 15 time steps |

[1] Negative represents opening and positive represents closing.

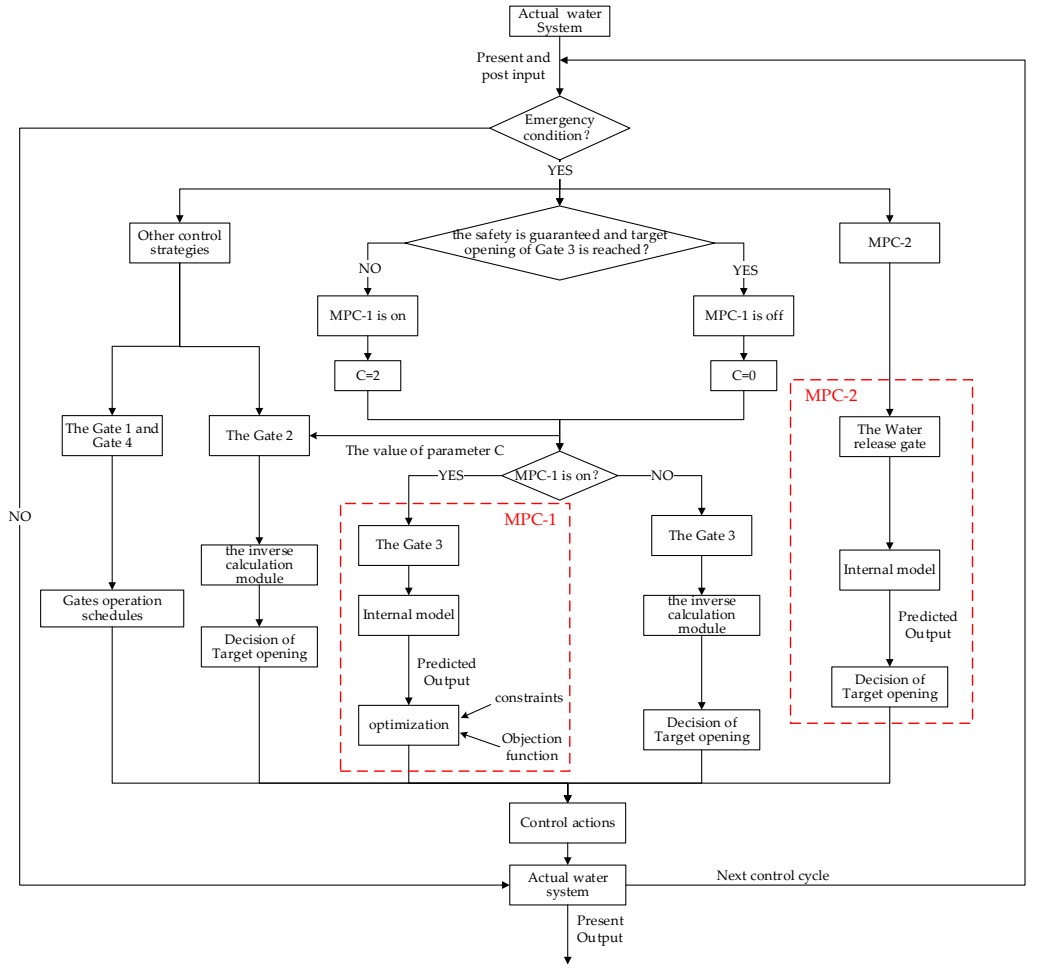

**Figure 3.** Schematic overview of the control system.

### 3.3.1. The Internal Model

1. The Linear Model for the Long Inverted Siphon

The authors recently proposed a linear model that related the pressure head variations at the downstream end of an inverted siphon (notated as $h_2(k)$) to the flow rate variations at two ends (notated as $q_1(k)$ and $q_2(k)$) [31]. It divided $h_2(k)$ in the inverted siphon into the low-frequency part and

high-frequency part, caused by the deformation of the siphon wall and the reflection of the acoustic wave, respectively. In Mao's study [31], the linear model was adopted to model two scenarios, one is a virtual large-scale inverted siphon and the other one is a PVC pipe. In the meantime, the accuracy of the linear model is verified in the frequency domain using the Bode plot, and the pressure head computed using the linear model is compared with the simulation results of finite volume method (FVM). The discrete time-invariant linear model for the long inverted siphon applied in this study is given in Equation (4).

$$h_2(k) = h_1(k-1) - h_f(k-1) - \frac{a}{Ag}[q_1(k-1) - q_1(k-1-k_d)] \tag{4}$$

where $k$ is time step; $h_1(k)$ and $h_2(k)$ are pressure head deviation at the upstream end and downstream end at time step $k$ (m), respectively; $h_f$ is the frictional head loss computed using the flow rate at the upstream end of the inverted siphon under the assumption of a quasi-steady flow condition (m); $a$ is the speed of the acoustic wave, which is taken as 1054 m/s; $A$ is the wetted area (m$^2$); $g$ is the acceleration of gravity (m/s$^2$); $k_d$ is the delay steps of the acoustic wave traveling from the upstream end to the downstream end, and it can be computed approximately using:

$$k_d = \frac{L}{a \times \Delta t} \tag{5}$$

where $L$ is the distance of the long inverted siphon (m); and $\Delta t$ is the computation time step, which is taken as 1 min.

In order to further verify the linear model, it is applied to the project as described in Section 2. The test scenario is given in Table 3 and the result is shown in Figure 4. In addition, the four-point difference implicit scheme of Pressman (i.e., finite difference method, FDM) is also used to solve the Saint-Venant equation. It can be seen from the Figure 4 that the results of the two methods are quite different during the gate operations, but on the whole, the linear model can reasonably reflect the unsteady flow characteristics at the downstream end of the inverted siphon (MAPE is mean absolute percentage error, and the smaller the value is, the better the effect of the model is). The reason for the difference is that there are many assumptions and simplifications in the derivation of the linear model, but it is generally acceptable.

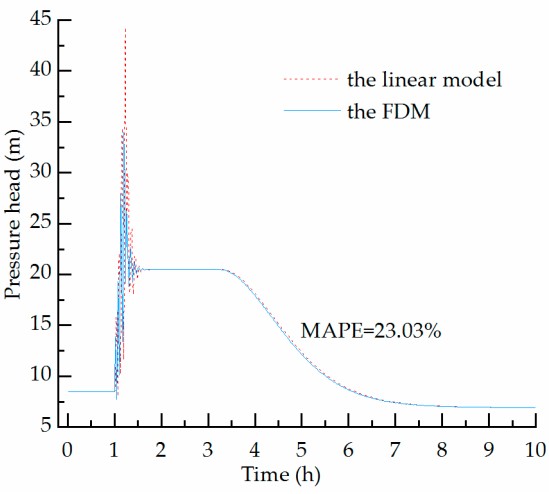

**Figure 4.** The verification to linear model.

**Table 3.** Gates schedule of the test scenario.

| Gate Number, $j$ | Gate Width, $b_j$ (m) | Gate Initial Openings, $e_{1j}$ (m) | Gate Target Openings, $e_{2j}$ (m) | Action Start Time, $T_{1j}$ (h) | Action end Time, $T_{2j}$ (h) |
|---|---|---|---|---|---|
| 1 | 7.4 | 3.92 | 0.95 | 1.5 | 1.75 |
| 2 | $5 \times 3$ | 3.84 | 0.93 | 1.5 | 1.75 |
| 3 | $3.8 \times 3$ | 3.8 | 0.92 | 1 | 1.25 |
| 4 | 8 | 3.84 | 0.93 | 1 | 1.25 |

## 2. The Linear Model for the Open Canal

For the pipe-canal combination system shown in Figure 1, the above linear model can be used as the internal model for the prediction to the long inverted siphon, and the integrator-delay (ID) model is commonly used for open canal [30]. It assumes that an open canal reach is separated into a uniform flow and a backwater section, as presented in Figure 5. Delay time ($\tau$ in s) and average storage area ($A_s$ in m$^2$) are the two main properties of each open canal reach in ID model, as can be seen in Table 4. The discrete time-invariant ID model applied in this study is:

$$H_d(k+1) = H_d(k) + \frac{\Delta t}{A_s}\left[Q_{in}(k-k_d) - Q_{out}(k) - Q_{off-take}(k)\right] \tag{6}$$

where $H_d(k)$ is the water depth at the downstream end of the pool at time step $k$ (m); $Q_{out}(k)$ is the control flow to the downstream reach at time step $k$ (m$^3$/s); $Q_{in}(k)$ is the inflow (m$^3$/s); $Q_{in}(k-k_d)$ is the inflow (m$^3$/s) to the backwater section with $k_d$ being the delay time step between control action and the change in average downstream water level; and $Q_{off-take}(k)$ is the off-take outflow which originates from the control of the water release gate (m$^3$/s).

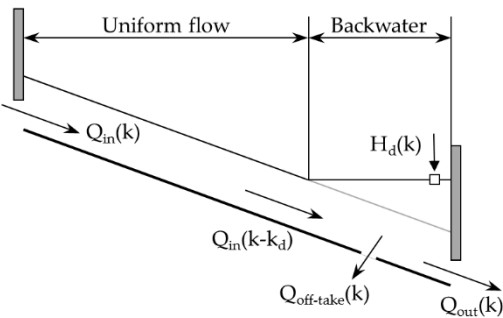

**Figure 5.** Schematization of a modeled canal reach by the integral delay (ID) model.

**Table 4.** Parameters of the integral delay (ID) model.

| Pool | Storage Area $A_s$ (m$^2$) | Delay Time $\tau$ (s) |
|---|---|---|
| 1 | 3,3536.12 | 420 |
| 3 | 7055.83 | 20 |

### 3.3.2. Receding Horizon

The receding horizon used in this paper is different from the conventional MPC [14]. A control cycle is completed every ten time steps, which means it is only once every ten minutes that the gates' actions are judged by MPC-1 and MPC-2. Accordingly, the prediction horizon is also set as ten time steps. This is done to prevent frequent opening and closing of gates and to reduce simulation time.

### 3.3.3. Constraints

MPC has the ability to deal with constraints. Controlled variable such as internal water pressure at the downstream end of the long inverted siphon is not allowed to violate their safety limit (between 3.8 m and 61 m) anytime within the prediction horizon, which can be described as:

$$3.8 \leq H_d(k+i) \leq 61 \tag{7}$$

where $i$ is the number of predicted steps, from 1 to 10; $k$ is the time step when MPC-1 is invoked every control cycle; $H_d(k+i)$ is the pressure head at the downstream end of the long inverted siphon, which is predicted at step $k+i$ (m).

### 3.3.4. Objection Function

In every control cycle, the future sequence over the prediction horizon of control actions will be optimized. This is done by minimizing an objective function with penalties on the pressure head deviations from setpoint which is taken as $H_d(k)$ and on the control options for gate operation (i.e., the effort that has to be put in controlling Gate 3). The objective function for MPC-1 is:

$$\min J = G \times NISE + W \times \Delta e_j \tag{8}$$

$$NISE = \frac{\frac{1}{10} \times \sum\limits_{i=1}^{n} [H_d(k+i) - H_d(k)]^2}{H_d(k)^2} \tag{9}$$

where $J$ represents the objective function that needs to be minimized; $\Delta e_j$ is the gate opening increment of option $j$; $NISE$ is the non-dimensional integrated square of error which can measure the stability of water level control [33]; $n$ is the number of steps over the prediction horizon; $H_d(k)$ is the pressure head in the downstream end of the long inverted siphon at time step $k$ (m); $G$ is the penalty on $NISE$ and the value is taken as 10. $W$ is the penalty on gate opening increment and is taken as Equation (10) to reduce the number of gate movements. The values of $G$ and $W$ are determined through trial and error.

$$W = \begin{cases} \frac{1}{\Delta e_j} & \Delta e_j = 0 \\ \frac{10}{\Delta e_j} & \Delta e_j \neq 0 \end{cases} \tag{10}$$

The MPC-2 on the water release gate does not contain the optimization process by minimizing the objective function. Instead, the target opening of the water release gate is determined just according to the prediction result of the last step in the prediction horizon every control cycle, as shown in Equation (11).

$$e_{target\_release} = \begin{cases} 0 & \Delta H_{gate2}(k+n) < 0.5 \\ 2 & 0.5 \leq \Delta H_{gate2}(k+n) < 0.8 \\ 4 & \Delta H_{gate2}(k+n) \geq 0.8 \end{cases} \tag{11}$$

where $e_{target\_release}$ is the target opening of the water release gate (m); $\Delta H_{gate2}(k+n)$ is the predicted deviation of water level compared with initial water level in front of Gate 2 at the last step of the prediction horizon (m).

If MPC-1 is invoked repeatedly throughout the simulation, it may cause frequent vibration of Gate 3. So it is set that MPC-1 will be shut down permanently from the moment when the pressure head has been within the safe range in the past one hour after the accident and Gate 3 has reached target opening. Due to the large difference between the wave speed of the open flow and the pressurized flow, there is a large hysteresis in open canals. Because of this, MPC-2 needs to be repeated through the whole process of simulation to prevent Pool 1 from overtopping.

### 3.4. Boundary Conditions and Other Control Strategies

The setting of boundary conditions has a great influence on the simulation results. The three pools together are treated as an independent canal in this study. The upstream boundary is set as the flow through Gate 1 which is assumed to vary linearly with the gate opening. The downstream boundary is similarly set as the flow rate of Gate 4, but computed by the free discharge formula of the sluice hole [34], as shown in Equation (12). Setting the boundary conditions in this way means the change progress of the flow through the Gate 1 is given, which may result in fewer operations of the water release gate because of less water delivered from upstream. However, it has little impact on the control and safety of the long inverted siphon.

$$\begin{cases} Q = \mu_0 eb\sqrt{2gH_0} \\ \mu_0 = 0.60 - 0.18 \times e/H_0 \end{cases} \tag{12}$$

where $e$ is the opening of the gate (m); $b$ is the width of the gate (m); $H_0$ is the water depth in front of the gate; $\mu_0$ is the coefficient of discharge.

In the model, the flow of Gate 2 and the water release gate is also computed by Equation (12) because of the free drop behind the gate. The flow of Gate 3 is submerged flow and is computed by the sluicegate discharge equation proposed by Henry, H R [35] and Swamee, P K [36]:

$$\begin{cases} Q = C_d be\sqrt{2gH_0} \\ C_d = 0.611\left(\frac{H_0-e}{H_0+15e}\right)^{0.072}(H_0 - H_2)^{0.7}\left\{0.32\left[0.81H_2\left(\frac{H_2}{e}\right)^{0.72} - H_0\right]^{0.7} + (H_0 - H_2)^{0.7}\right\}^{-1} \end{cases} \tag{13}$$

where $C_d$ is the coefficient of the sluicegate discharge; $H_2$ is the water depth behind the gate (m). For overflow weir, its flow can be computed by free-overflow formula of wide crested weir [34]:

$$Q = \delta_s mb\sqrt{2g}H_0^{3/2} \tag{14}$$

where $\delta_s$ is the coefficient of side-contract and is taken as 0.9; $m$ is the coefficient of discharge and is taken as 0.32; $b_{weir}$ is the width of weir crest (m); $H_0$ is the depth in front of the overflow weir, m.

For quick adjustment, the opening of Gate 2 is determined every 15 min by the inverse calculation module of gate opening for a given target flow, which is decided as the following:

$$Q_{22} = Q_2 + C \tag{15}$$

$$C = \begin{cases} 2 & MPC - 1 = 1 \\ 0 & MPC - 1 = 0 \end{cases} \tag{16}$$

where $Q_2$ is the target steady-state flow rate for emergency dispatch (m³/s); $Q_{22}$ is the target flow of Gate 2 for inverse calculation module. When the sudden accident occurs, the inflow of Pool 2 can be quickly adjusted to target flow, but the outflow of Pool 2 is controlled by MPC-1 which suppresses the rapid changes of flow to prevent pressure surge. Inevitably, there is a tendency of storage decreasing in Pool 2 and it is necessary to replenish the incoming water to prevent the inverted siphon intake, which is why the parameter C is set.

As for Gate 3, the exit gate of the long inverted siphon, in order to adjust the gate flow to the target flow as soon as possible and prevent frequent gate action, it is also controlled every 15 min by inverse calculation module of gate opening after the MPC-1 is shut down. In addition, Gate 1 and Gate 4 are operated according to the gates schedule for simplicity.

### 3.5. Test Scenarios

Here, the main goal is to ensure the control effect of MPC when there is a sudden flow obstruction incident in the downstream of the Menglou~Qifang inverted siphon. To this end, four test scenarios are chosen according to the risk of the accident, as can be seen in Table 5.

According to the target flow in each scenario, the steady flow calculation is carried out, and the target opening of Gate 3 (the setting parameter of MPC-1) is preliminarily obtained. The target opening of Gate 1 and Gate 4 is simply determined by Equation (17):

$$e_{2j} = e_{1j} \times \frac{Q_{2j}}{Q_{1j}} \tag{17}$$

where $j$ is the number of the gate; $e_{1j}$ is the initial opening of gate $j$ (m); $e_{2j}$ is the target opening of gate $j$ (m); $Q_{1j}$ is the initial flow of gate $j$ (m³/s); $Q_{2j}$ is the target flow of gate $j$ (m³/s).

It is assumed that an accident occurs when T = 1 h and emergency control is activated. The upstream gate (Gate 1) and downstream gate (Gate 4) operate according to the schedules in Table 5, and the middle gates (the water release gate, Gate 2, and Gate 3) are automatically controlled by the algorithm/strategies presented in Sections 3.3 and 3.4.

**Table 5.** Control parameters of test scenarios.

| Scenario | A | B | C | D |
|---|---|---|---|---|
| Accident risk | Low | Moderate | Serious | Huge |
| Initial flow, $Q_1$ (m³/s) | | 38 | | |
| Target flow, $Q_2$ (m³/s) | 30 | 20 | 10 | 0 |
| Initial opening of the Gate 1, $e_{11}$ (m) | | 3.92 | | |
| Target opening of the Gate 1, $e_{21}$ (m) | 3.09 | 2.06 | 1.03 | 0.00 |
| Action start time of the Gate 1, $T_{11}$ (h) | | 1.5 | | |
| Action end time of the Gate 1, $T_{12}$ (h) | | 1.75 | | |
| Target opening of the Gate 3 for MPC-1, $e_{23}$ (m) | 2.60 | 1.50 | 0.90 | 0.0 |
| Initial opening of the Gate 4, $e_{14}$ (m) | | 3.84 | | |
| Target opening of the Gate 4, $e_{24}$ (m) | 3.03 | 2.02 | 1.01 | 0.00 |
| Action start time of the Gate 4, $T_{14}$ (h) | | 1 | | |
| Action end time of the Gate 4, $T_{24}$ (h) | | 1.25 | | |

## 4. The Simulation Results

The study about emergency control carried out in this paper pays more attention to the safety of the project, which is guaranteed by MPC, and there is no offtake along the line, so there is no need to maintaining a constant water level at some point like other researches about canal automation [37]. The results of the simulation are shown in Figures 6–9, which show that the proposed control algorithm and strategies can fulfill the emergency control tasks automatically under different risks of flow obstruction incidents downstream.

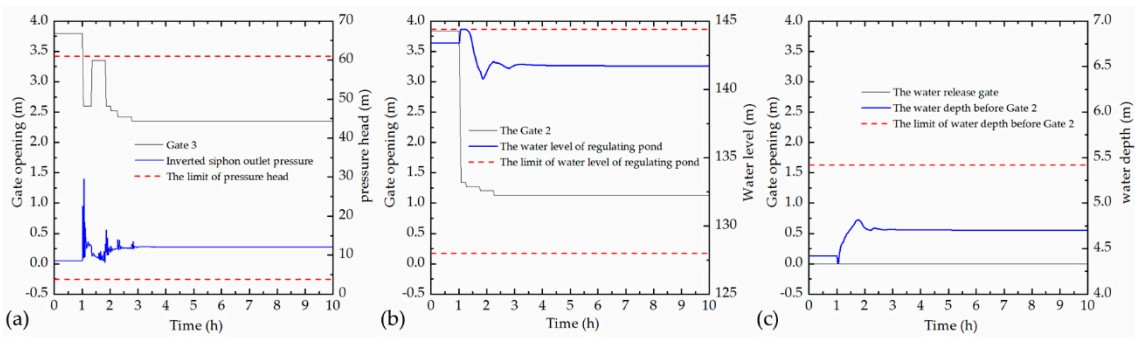

**Figure 6.** The simulation result of Scenario A with target flow of 30m³/s: (**a**) the trajectories of Gate 3 and the change of pressure at inverted siphon outlet; (**b**) the trajectories of Gate 2 and the change of water level in regulating pond; (**c**) the trajectories of the water release gate and the change of water depth before Gate 2.

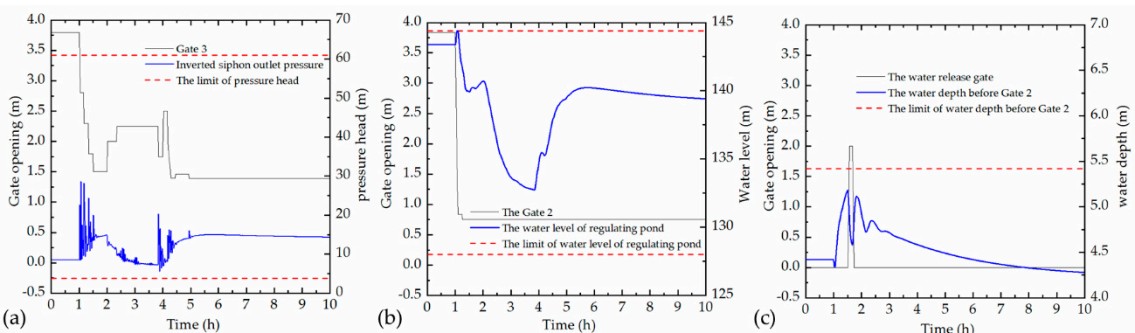

**Figure 7.** The simulation result of Scenario B with target flow of 20 m³/s: (**a–c**) are same with Figure 6.

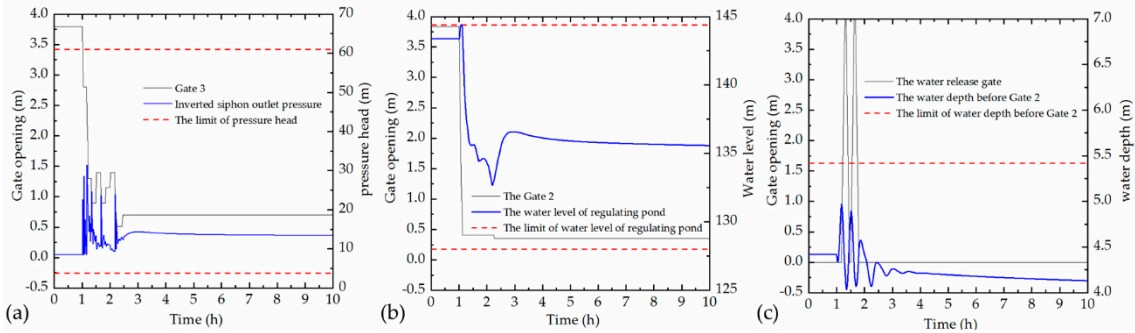

**Figure 8.** The simulation result of Scenario C with target flow of 10 m³/s: (**a–c**) are same with Figure 6.

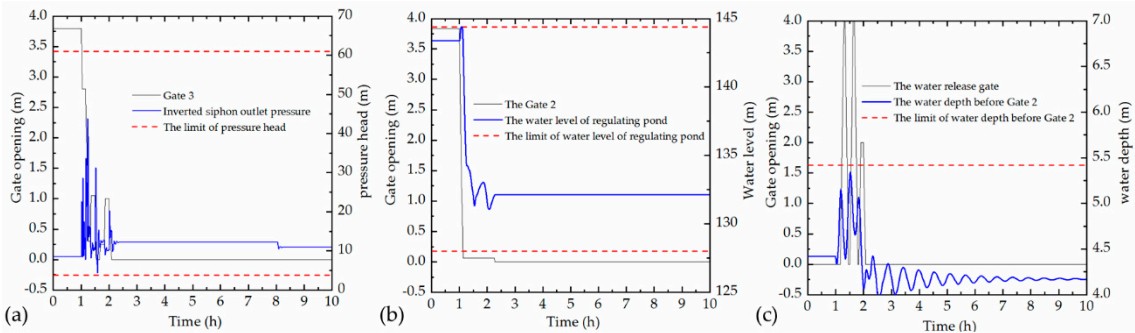

**Figure 9.** The simulation result of Scenario D with target flow of 0 m³/s: (**a–c**) are same with Figure 6.

The detailed simulation results of these four test scenarios are provided in Table 6, from which some trends can be seen:

1.  NIAW is non-dimensional integrated absolute gate movement, which can measure the opening and closing amplitude and frequency of the gate [33], and can be calculated by Equation (18). NIAW of Gate 3 and maximum pressure head in front of Gate 3 are on the rise with the risk of accident increasing, except for Scenario B, which will be analyzed later. It means that the condition is becoming more and more difficult to control from Scenarios A to D. For the safety of long inverted siphon, Gate 3 has to go through a more complex process of opening and closing determined by MPC-1. In addition, the NIAW of Gate 2 is zero under all scenarios, indicating that Gate 2 is gradually closed in the progress of automatic control without being opened and closed frequently, as can be seen in Figures 6b, 7b, 8b and 9b.

$$NIAW = \frac{\frac{\Delta t}{T}\left(\sum\limits_{t=t_1}^{t_2}|e_t - e_{t-\Delta t}| - (e_{t_1} - e_{t_2})\right)}{e_{max}} \tag{18}$$

where $t$ is time (min); $t_1$ and $t_2$ are the moments for flow changing and stabilizing (min); $\Delta t$ is the discrete-time step of control systems, which is taken as 1 min; $T$ is the simulation time (min); $e_t$ is the gate opening at time $t$ (m); $e_{max}$ is the maximum gate opening (m).

2.　There is an obvious increase in the NIAW of the water release gate and water abandoned from Scenarios A to D. When Gate 2 is quickly closed by the inverse calculation module, the water level in front of Gate 2 rises rapidly. The MPC-2 shows good performance on predicting the change of water level and controlling the water release gate ahead of time to prevent the overtopping accident. In consequence, the more serious the backwater caused by Gate 2 is, the more frequent the water release gate action is. Scenario D is the most urgent condition, and it can be seen in Figure 9c that the water release gate has been opened and closed for a total of three times to deal with the sharp fluctuation of the water level before Gate 2. In the meantime, the overflow weir also works but not in Scenarios A to C. According to the schedule for Gate 4, it will be completely shut down to prevent the water from flowing downstream and causing a secondary accident. However, at this time, Gate 3 is not completely closed under the control of MPC-1 for the safety of the long inverted siphon and continued to discharge into the Pool 3 for a period time, which leads to the rapid rise of water level in the canal Pool 3 and overflow. It can be seen in Figure 10 that the overflow lasts about 1.5 h, and the water level in Pool 3 finally stabilizes at the elevation of the overflow weir crest, 128.5 m.

3.　The water level in regulating pond is determined by the variation of the flow through Gate 2 and the flow into the long inverted siphon. Due to the fast velocity of the acoustic wave, the latter is greatly affected by the Gate 3. The simulation results show that there is a drop in the stable water level in regulating pond from Scenarios A to D. When there is an accident, the flow of Gate 2 can be quickly adjusted to the target flow through the inverse calculation module, but the downstream outflow of the long inverted siphon still needs a period time to adjust (judged by MPC-1). The continuous flow difference between upstream and downstream leads to the decline of the water level in the regulating pond. With the increase of accident risk, the flow difference between the upstream and downstream of Pool 2 is greater, which results in a decrease in the stable water level in the regulating pond. Furthermore, a sharp increase of water level can be seen in the regulating pond when Gate 2 and Gate 3 begin to move. There are several possible explanations for this, but the main reason may be the influence of water hammer waves. Gate 3 is located at the junction of pressurized flow and open flow, and closing with maximum speed when the accident occurs will result in severe water hammer. The rapid propagation of the water hammer wave upstream leads to a sharp decrease in the flow into the long inverted siphon, which is larger than the decrease of flow of Gate 2 due to the hysteresis of the open flow. Consequently, the water level in the regulating pond rises rapidly at the beginning of the accident. But it will decrease soon with the closing of Gate 2 and the reflection of the water hammer wave by the regulating pond.

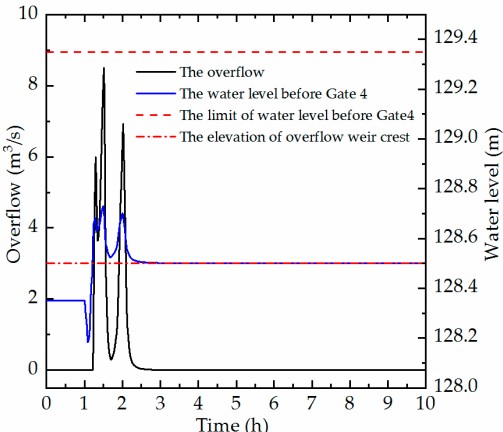

**Figure 10.** The overflow and water level in Pool 3.

**Table 6.** Simulation results of test scenarios.

| Scenario | A | B | C | D |
|---|---|---|---|---|
| *NIAW* of Gate 3 | $1.97 \times 10^4$ | $4.11 \times 10^4$ | $3.13 \times 10^4$ | $4.61 \times 10^4$ |
| *NIAW* of Gate 2 | 0 | 0 | 0 | 0 |
| *NIAW* of the water release gate | 0 | $0.53 \times 10^3$ | $2.11 \times 10^3$ | $2.63 \times 10^3$ |
| The volumes of water release (m$^3$) | 0 | 18,598 | 63,504 | 85,878 |
| The volumes of overflowing (m$^3$) | 0 | 0 | 0 | 11,659 |
| Initial water level of regulating pond (m) | | | 143.4 | |
| Stable water level of regulating pond (m) | 141.7 | 139 | 135.3 | 132.1 |
| Initial pressure head in front of Gate 3 (m) | | | 8.6 | |
| Maximum pressure head in front of Gate 3 (m) | 29.6 | 28.7 | 31.5 | 43.8 |
| Upper limit of pressure head (m) | | | 61 | |
| Initial water depth in front of Gate 2 (m) | | | 4.42 | |
| Maximum water depth in front of Gate 2 (m) | 4.81 | 5.19 | 4.97 | 5.35 |
| Upper limit of water depth (m) | | | 5.42 | |

The simulation result of Scenario B does not conform to the overall trend, which is very likely to be related to the MPC parameters settings and the low prediction accuracy of the linear model during the gate operation. The latter will be discussed in Section 5. These four scenarios designed in Table 5 are all simulated by the same program without modifying any setting parameters, such as the prediction horizon, the control interval, the cost-weighting matrix, or the control action options for gate opening increment. Because of this, it is possible that for each scenario, the setting of the control parameters is not the most optimal, which shows most obvious in Scenario B, but it is acceptable if the control purposes can be achieved. Moreover, it may be more suitable for engineering practice to deal with different levels of flow obstruction accidents with the same set of parameters.

To further study the effect of MPC-1 and MPC-2, two control groups are set up based on the Scenario D. One (Scenario D-1) is simulated without MPC-1, the other one (Scenario D-2) is simulated without both MPC-1 and MPC-2, and the results are shown in Figures 11 and 12. Looking at Figure 11, it is apparent that the pressure in the inverted siphon is at the critical point of exceeding the limit, which is not accepted, and the overtopping accident is also easy to occur in front of the Gate 2. Without the control of MPC-1, Gate 3 is shut down with the maximum speed (0.5 m/min), accompanied by severe water hammer which can lead to the burst of inverted siphon. Comparing Figures 9a and 11a, it can be seen that the MPC-1 based on the linear model for long inverted siphon [31] can successfully predict the peak pressure and adjust the action of Gate 3 ahead of time to avoid danger. When neither MPC-1 nor MPC-2 is enabled, not only the inverted siphon may burst, but also a serious overtopping accident will occur in Pool 1, just as shown in Figure 12. As for the regulating pond, the water level did not drop to a relatively low level as shown in Figure 9b. It is mainly because that the MPC-1 was closed and the gates at both ends were closed quickly so that the water was fully stored in Pool 2, rather than flowing downstream. Besides, the water level in the regulating pond is finally maintained at a value slightly higher than the initial water level. It is mainly related to the non-synchronous change of flow during the closing of Gate 2 and Gate 3.

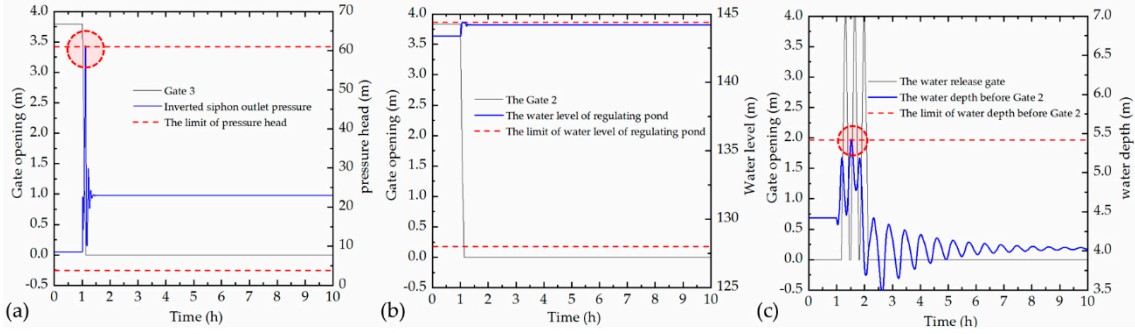

**Figure 11.** The simulation result of Scenario D-1 without MPC-1: (**a**–**c**) are same with Figure 6.

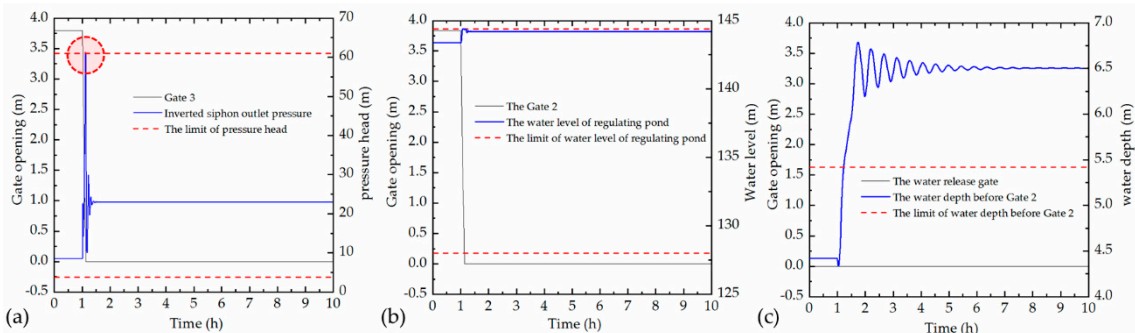

**Figure 12.** The simulation result of Scenario D-2 without MPC-1 and MPC-2: (**a–c**) are same with Figure 6.

Scenario D-2 is actually a typical and conventional emergency dispatching plan, in which Gate 1~Gate 4 are closed as quickly as possible to prevent water from flowing downstream. According to similar engineering reports, the operators are likely to control the gates in this way when a huge downstream accident occurs and there is no similar engineering experience for reference. It can be seen from Figure 12 that this emergency control effect is extremely poor. Of course, it is possible to obtain a set of safe results by continuously adjusting gate trajectories plan, but when another level of accident occurs, the same work needs to be repeated. Compared with advanced model predictive control, the traditional method is quite time-consuming, laborious and inefficient.

## 5. Discussion

This paper takes the Northern Hubei Water Transfer Project as the research object and takes MPC as the control method to study the emergency dispatching. Although this project is particular due to the 72 km long inverted siphon, it can also fully illustrate the advantages and applicability of MPC in emergency control. According to the results in Section 4, it is clear that MPC has good performance in predicting the fluctuation of water level and pressure head to prevent secondary accidents. Nevertheless, there are also some aspects worthy of attention and discussion.

First of all, the selected speed of the acoustic wave is an important factor affecting simulation results. Malekpour and Karney [38] investigated the source of the spurious numerical oscillations often observed in simulations using the well-known Preissmann slot method (PSM). They pointed out that PSM cannot sustain the negative pressures that frequently occur in simulations and spurious numerical oscillations are often induced when the flow switches from the open canal to pressurized flow with the higher acoustic wave velocities being introduced. In other words, when the real acoustic wave velocity is used, which is likely higher than 1054 m/s and usually needs to be measured by experiment, the pressure oscillation before the Gate 3 may lead to temporary negative pressure, frequently prematurely terminating the simulation. Three-dimensional simulation or finite volume method can better solve this problem, but for the long inverted siphon in this paper, it will consume a lot of computer resources, which makes the tuning and application extremely difficult.

After this, how to choose the boundary conditions is also influential, especially for the exit gate (Gate 3) of the long inverted siphon, which has a significant impact on the hydraulic response process in the pipes. Gate 3 is located at the junction of the pressurized section and the open canal section, belonging to the orifice-submerged outflow of the long pressurized pipeline, which is highly coupled and non-linear. However, there is little research on this kind of complex boundary. The sluicegate discharge equation proposed by Henry [35], which is selected as the boundary condition for the Gate 3, has been verified by many scholars for many years as a classical empirical formula for calculating gate discharge. But it is more suitable for the gates at the reservoirs or dams [39]. Whether it is suitable for the exit gate of the long inverted siphon needs to be further verified by prototype-observation or three-dimensional simulation. In future research, it might be possible to deal with the above two problems by coupling 1D and local 3D simulation. More specifically, a small region surrounding the

moving exit gate of the long inverted siphon is resolved by computational fluid dynamics (CFD), using a dynamic mesh library, while the rest of the system is modeled by FDM. Furthermore, the volume of fluid (VOF) algorithm is needed in CFD to simulate the two-phase flow patterns for its advantageous ability of tracking the gas–liquid interface. The applications of similar collaborative simulation technology show its feasibility [40,41], but it is rarely used in the large-scale pipe-canal combination system and can be tried in the further studies.

Along the line of consideration, there is another factor that has an impact on the accuracy of the simulation results, and that is the value of computation time step $\Delta t$. The smaller the $\Delta t$ is, the greater the peak pressure in the pipes can be captured during the simulation, but the longer the simulation time is. Based on the comprehensive consideration, $\Delta t$ is taken as 1 min in this paper for the Northern Hubei Water Transfer Project. It could be a good choice to use different computation time steps in the open flow section and the pressurized section for more accurate results.

Last but not least, the linear model of water movements for large-scale inverted siphon [31] is not perfect. Despite the linear model has good performance in MPC-1, there are still some gate operations that may be unnecessary for guaranteeing the safety of the long inverted siphon, such as the gate operations between 2 h and 5 h shown in Figure 7a. One reason for this may be that during the time of gate operation, the prediction result of the linear model fluctuates more violently than the actual pressure (as shown in Figure 4), so that although it can successfully predict the time when the pressure peak occurs and weaken the pressure peak ahead of time by MPC-1, it will also lead to frequent movements caused by misjudgment. Further research on this linear model is a solution for more precise control.

## 6. Conclusions

Based on the linear model for large-scale inverted siphon proposed by Mao et al. [31], the present study is designed to evaluate the applicability of model predictive control in the case of an emergency. On the basis of the aforementioned results and analysis, the following conclusions can be drawn:

1. When there is no similar engineering experience for reference, the traditional method of making emergency dispatching plan is inefficient. If there is an accident that is not considered in the plans, the performance of emergency control may be greatly reduced, or even a secondary accident may occur due to the fast hydraulic transient characteristics of pressurized flow in the long inverted siphon. At this point, automatic control can be a good choice.

2. The MPC algorithm proposed in this paper can effectively prevent the long inverted siphon from bursting in the outlet section and overtopping in the inlet section when there are sudden flow obstruction incidents of varying degrees downstream. With the rise of accident risk, the control difficulty for MPC also increases, which can be reflected in more complicated gates operations (e.g., the NIAW of Gate 3 in huge risk scenario is more than twice as high as that in low risk scenario) and more water release (up to $9.75 \times 10^4$ m$^3$).

3. The predicted results of this linear model can help MPC to reduce the peak pressure by taking action ahead of time and ensure the safety of the project, such as the decrease in peak pressure by 17.2 m compared with the results of Scenarios D and D-1. However, due to the poor accuracy of the prediction results during the gate operation, it may lead to some unnecessary gate movements judged by MPC.

The Northern Hubei Water Transfer Project, which has a 72 km long inverted siphon, is a typical and rare pipe-canal combination system. The great difference between the wave velocity of open flow and pressurized flow makes the control of this kind of project, especially under accident conditions, extremely difficult. On the one hand, it is necessary to ensure the safety of hydraulic structures, on the other hand, excessive gate movements should be avoided. That is why a lot of time is spent on tuning the objective function and control parameters to adapt to different degrees of downstream flow obstruction incidents. This work can be a reference for the emergency dispatching of similar projects.

Further studies need to be carried out to improve the accuracy of the linear model and evaluate the feasibility of combining local 3D-simulation.

**Author Contributions:** Conceptualization, Z.Z., G.G., Z.M., K.W., S.G., G.C.; resources, Z.Z., G.G., Z.M., K.W., S.G., G.C.; writing—original draft, Z.Z.; writing—review and editing, Z.Z. All authors have read and agreed to the published version of the manuscript.

**Funding:** The authors acknowledge the support of the NSFC grant 51979202 and NSFC grant 51009108.

**Conflicts of Interest:** The authors declare no conflict of interest.

## Appendix A

### Nomenclature

| Notation | Definition | Units |
|---|---|---|
| $k$ | the simulation step | \ |
| $\triangle t$ | computation time step | min |
| $T$ | the simulation time | min |
| $T_{1j}$, $T_{2j}$ | the action start and end moment of gate j | h |
| $t_1$, $t_2$ | the moments for flow changing and stabilizing | min |
| $\tau$, $k_d$ | the delay time and delay time steps | min, \ |
| $a$ | the speed of the acoustic wave | m/s |
| $g$ | the acceleration of gravity | m/s$^2$ |
| $Q$, $v$ | the discharge/ flow velocity | m$^3$/s, m/s |
| $Q_{in}(k)$, $Q_{out}(k)$, $Q_{off\text{-}take}(k)$ | the inflow/outflow/off-take at step k | m$^3$/s |
| $e$, $\Delta e$ | the gate opening and increment | m |
| $e_t$, $e_{max}$ | the gate opening at time t and the maximum gate opening | m |
| $e_{1j}$, $e_{2j}$, $Q_{1j}$, $Q_{2j}$ | the initial and target opening/flow of gate j | m, m$^3$/s |
| $e_{target\_release}$ | the target opening of the water release gate | m |
| $q_1$, $q_2$ | the relative discharge at the upstream/ downstream end of the long inverted siphon | m$^3$/s |
| $H$ | the water depth in the open flow, or pressure head in the pressurized flow | m |
| $H_0$, $H_2$ | the water depth before and after the gate/weir | m |
| $H_d(k)$ | the water level/ the pressure head at the downstream end of the open pool or inverted siphon at step k | m |
| $h_1(k)$, $h_2(k)$ | the pressure head deviation at the upstream/ downstream end of the long inverted siphon | m |
| $h_f$ | the frictional head loss | m |
| $\Delta H_{gate2}(k)$ | the predicted deviation of water level compared with initial water level in front of Gate 2 | m |
| $A$, $A_s$ | the wetted area/ storage area | m$^2$ |
| $B_{sl}$, $B$ | the width of the narrow slot, and the width of the water surface in the open flow, or the width of slot in the pressurized flow | m |
| $b$, $b_{weir}$ | the width of gate/ the weir crest | m |
| $L$ | the distance of the long inverted siphon | m |
| $G$, $W$, $C$ | the plenty coefficient and the flow for replenishing | \, m$^3$/s |
| $m$, $C_d$, $\mu_0$, $\delta_s$ | the coefficient of discharge/ side -contract | \ |
| $S_t$, $S_b$ | the hydraulic slope/ bed slope | \ |
| $n$, $R$ | the Manning coefficient/ hydraulic radius | \, m |
| $J$ | the objective function | \ |

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
