# Peer review of "Application of Model Predictive Control for Large-Scale Inverted Siphon in Water Distribution System in the Case of Emergency Operation"

_water, doi:10.3390/w12102733_

Round 1
Reviewer 1 Report
In this research, a MPC approach with a linear model was proposed to optimize the operation of the inverted siphon in Hubei China for emergency operation. Overall, the paper is well written in terms of objective, method, and the interpretation of results. Several comments are given as below for improvement before accepted for publication:
- An overall proofread is suggested to fix some grammar and writing errors. For example, “for constraints to allowable” in L71; “sluice hole be used for” in L112, “siphon intake or overtopping” in L120, and so on.
- Please provide the full definitions of some abbreviations when literatures are reviewed. For example, the “GoRoSo” in L54; the “PI” in L59, and so on.
- The sentence “so there is no need to worry” in L122 seems to be too colloquial.
- In Figure 2, the linear model predicts the peak of pressure head much higher than the FDM. Will this result in a rather conservative gate operation? Please give comments on it.
- The expressions of the governing equations in Eq. (4) are too simplified. Some important details such the determination of H are missing. In addition, the term “A/Bsl” in the continuity equation may not represent the whole cross-sectional area of water.
- The openings of Gate 1 and Gate 4 are determined simply by linear relationship with discharges in Eq. (16). I am not sure this is theoretically correct because the relationships between discharge “Q” and opening “e” are not linear according to Eq. (11) and Eq. (12). Please discuss more on how these assumptions affect the results.
- In L326, the statement “an obvious increase in the NIAW ... from Scenario A to D” is not correct; at least not stand for Scenario B.
- In L360-367, the deviation of Scenario B’s simulation results are attributed to the MPC parameter setting, which is either true or false without proof. Sensitivity analysis of the parameters will be helpful.
- In the conclusions, the authors argued that a 3D model will be helpful to deal with the negative pressure. However, there are tons of 3D model but not all are appropriate. More details should be given in terms of the requirements a 3D model should meet to serve the purpose.
Reviewer 2 Report
Writing and organization of the paper must be improved. Selection of MPC control must be justified and advantages with respect to conventional methods should be discussed.
Introduction:
The problem statement is not clear. Then it is difficult to understand which variables are involved in emergency control in canal systems, what are the control decisions and which is the benefit of using MPC for emergency control in canal systems. Moreover, the application to NHWTP is not clear.
-Northern Hubei Water Transfer project(NHWTP), which is about 72kmà project (NHWTP), 72 km (Uppercase and spaces)
Use brackets or authors name for citations between lines 53-89: Soler et al. (2013)[9], Lian et al. (2013)[10], Xu et al. (2017)[11]…
In my opinion these ideas are contradictory, first it is said that: “... few studies of MPC focus on emergency control of canal systems. Vierstra(2010)[24] dealt with an unexpected failure of a pump station in the South-North water transfer project with a maximum deviation of 0.36 meters. (How MPC was used? Is there any advantage?)” And in the next paragraph, said: “In general, MPC is used in the open canal system, rather than the pressurized flow system. The main reason is that there are mature linear models for open canal flow as…”. The second phase suggest that exist previous works using MPC for open canal systems control, and mature models are available, but in the previous paragraph it is mentioned that few works have used MPC technology for canal systems control.
Then, writing should be revised to clarify it and such previous works should be cited.
Writing should be revised in general.
In section 2:
the Northern Hubei water transfer projectà The Northern Hubei …
I don’t understand this phase: “In order to study the hydraulic response of Menglou~Qifang inverted siphon under accident conditions, the research scope of this paper should be larger than the long inverted siphon by extending an open canal at each end (station 20+200 ~ 97+920) which covers 3 pools. The initial flow rate is the design flow rate, 38m3/s, and there are no offtakes along the research scope.”
It would be interesting a brief description of typical operation conditions and possible causes of accidents in the water system.
Using the expression: “design scope” between lines 97 and 100 it is not the best option to describe system limits. It is confusing.
Section 2 (the Northern Hubei water transfer project) and Section 3 (Methods): It is difficult to associate process description with control problem and model variables. Both sections should be improved including an explicit description of system variables. A list of variables is necessary.
Section 3 (Methods):
Control actions for emergency operation are presented, but description is vague and difficult to understand and associate with system variables and control strategy. It should be improved.
Control problem description must be clear, it is necessary to understand how MPC strategy is implemented and what are the advantages of MPC with respect to other methods applied to inverted siphon under emergency conditions.
In section 3.1, when describing the linear model, the notation used to identify variables (pressure head and flow variations) could be introduced in lines 138-139.
In Model predictive control section (3.3), an overview of control structure is necessary at the beginning of section.
MPC strategy is proposed to deal with constraints, but conventional control strategies are not mentioned or compared with proposed MPC strategy.
Reviewer 3 Report
The work presents an application of a Model Predictive Control problem in designing emergency operational scenarios in a large scale inverted siphon. The paper from engineering point of view is interesting however I can not see a scientific merit. The authors should add a clear statement for what new brings their algorithm, what were any previous methodologies applied on these type of problems etc. I would expected within introduction section a clear representation of previous related problem statement and how they handled. This discussion/comparison is missing and the topic is failing to an interesting engineering case study. I'm happy to review again the current study once scientific soundness will be presented.Author Response
Please see the attachment.

Round 2
Reviewer 2 Report
Abstract:
“…this paper proposed…”-àthis paper proposes
“…which is popular in dealing with…” -à revise language
“…by predicting.” à revise language
“The study highlights both the applicability of the linear model for large-scale inverted siphon and the MPC in emergency dispatching.” àrevise writing
Introduction:
“In these studies, the research object of emergency dispatching is the open water system, rather than the pipe-canal combination system, especially the NHWTP which has a super-long inverted siphon.”à Check writing, the last phrase (“especially the NHWP…”) does not correspond to the first statement.
-MAO—(Mao et al.?) please cite MAO appropriately.
-Authors point out the advantage of MPC dealing with constraints, but this characteristic is not taken into account in the paper objective, that focuses on the use of the linear model.
The Northern Hubei Water Transfer Project
-Line 93, the title begins with a lowercase “the”.
-These expressions: (station 20+200 ~ 97+920) (station 25+520 ~ 97+600) should be briefly explained and associated with figure 1.
- lines 167, 168: “control schemes” ? I think you mean: control actions or control decisions?
- please cite MAO
Section 3, section 4.
This section has been significantly improved, but I think it is necessary a better description of control system including MPC1 and MPC2. It could be necessary to include a block diagram.
The terms related with control strategy description are confusing and sometimes, they are incorrect (I think it is a language problem). Then, “control scheme” is used instead of control actions in some and the terms: control strategy, control scheme and control structure are not well employed.
“MPC is capable of foreseen delivery problems…” àrevise language
Line 191 and Table 2: … “ control structural”: It means formulation of the control problem?
It is a good idea to include information of Table 2, but it is still difficult to understand.
Now, I think that enough information is provided but writing, use of language and organization of these sections makes it difficult to understand how the proposed control strategy is employed and to analyze simulation results.
Then, writing and organization need to be improved.
Reviewer 3 Report
My comments have been fully addressed and the paper has been improved substantially. I recommend it for publishing.
Author Response
We thank the reviewer for the carefulness and effort spent on reviewing this paper, this drives us to refine the manuscript for a better work.